# Coresets via Bilevel Optimization for Continual Learning and Streaming

**Zalán Borsos**
Dept. of Computer Science
ETH Zurich
zalan.borsos@inf.ethz.ch

**Mojmír Mutný**
Dept. of Computer Science
ETH Zurich
mojmir.mutny@inf.ethz.ch

**Andreas Krause**
Dept. of Computer Science
ETH Zurich
krausea@ethz.ch

## Abstract

*Coresets* are small data summaries that are sufficient for model training. They can be maintained online, enabling efficient handling of large data streams under resource constraints. However, existing constructions are *limited* to *simple models* such as $k$-means and logistic regression. In this work, we propose a novel coreset construction via *cardinality-constrained bilevel optimization*. We show how our framework can efficiently generate coresets for deep neural networks, and demonstrate its empirical benefits in continual learning and in streaming settings.

## 1 Introduction

More and more applications rely on predictive models that are learnt online. A crucial, and in general open problem is to reliably maintain accurate models as data arrives over time. *Continual learning*, for example, refers to the setting where a learning algorithm is applied to a sequence of tasks, without the possibility of revisiting old tasks. In the *streaming* setting, the data arrives sequentially and the notion of task is not defined. For such practically important settings where data arrives in a non-iid manner, the performance of models can degrade arbitrarily. This is especially problematic in the non-convex setting of deep learning, where this phenomenon is referred to as *catastrophic forgetting* [42, 23].

One of the oldest and most efficient ways to combat catastrophic forgetting is the *replay memory*-based approach, where a small subset of past data is maintained and revisited during training. In this work, we investigate how to effectively generate and maintain such summaries via *coresets*, which are small, weighted subsets of the data. They have the property that a model trained on the coreset performs almost as well as when trained on the full dataset. Moreover, coresets can be effectively maintained over data streams, thus yielding an efficient way of handling massive datasets and streams.

**Contributions.** We present a novel and general coreset construction framework,[1] where we formulate the coreset selection as a *cardinality-constrained bilevel optimization* problem that we solve by greedy forward selection via matching pursuit. Using a reformulation via a proxy model, we show that our method is especially well suited for replay memory-based continual learning and streaming with neural networks. We demonstrate the effectiveness of our approach in an extensive empirical study. A demo of our method for streaming can be seen in Figure 1.

## 2 Related Work

**Continual Learning and Streaming.** Continual learning with neural networks has received an increasing interest recently. The approaches for alleviating catastrophic forgetting fall into three main categories: using weight regularization to restrict deviation from parameters learned on old tasks

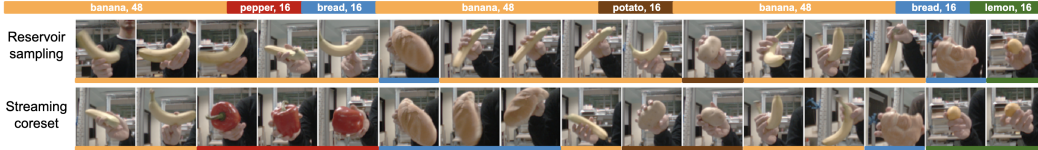

Figure 1: Data summarization on an imbalanced stream of images created from the iCub World 1.0 dataset [16]. First row: the stream's composition containing 5 object classes. Second row: selection by reservoir (uniform) sampling. Third row: selection by our method. Reservoir sampling misses classes (pepper) due to imbalance and does not choose diverse samples, in contrast to our method.

[32, 43]; architectural adaptations for the tasks [48]; and replay-based approaches, where samples from old tasks are either reproduced via a replay memory [39] or via generative models [50]. In this work, we focus on the replay-based approach, which provides strong empirical performance [9], despite its simplicity. In contrast, the more challenging setting of streaming using neural networks has received little attention. To the best of our knowledge, the replay-based approach to streaming has been tackled by [2, 27, 11], which we compare against experimentally.

**Coresets.** Several coreset definitions exist, each resulting in a different construction method. A popular approach is to require uniform approximation guarantees on the loss function, e.g., coresets for $k$-means [19], Gaussian mixture model [41] and logistic regression [29]. However, this approach produces a large coreset for more complex hypothesis classes, prohibiting its successful application to models such as neural networks. Several approaches relax the uniform approximation guarantees: the Hilbert coreset [7] which poses the subset selection problem as a vector approximation in a normed vector space, and data selection techniques that were successfully employed in active learning with convolutional neural networks [56, 49].[2] The term "coreset" in the continual learning literature is used more loosely and in most cases it is a synonym for the samples kept in the replay memory [43, 17], with the most common construction methods being uniform sampling and clustering in feature or embedding space. While in [27] the authors compare several data selection methods on streaming, to the best of our knowledge, no thorough analysis has been conducted on the effect of the data summarization strategy in replay memory-based continual learning, which we address with an extensive experimental study.

**Bilevel optimization.** Modeling hierarchical decision making processes [54], bilevel optimization has witnessed an increasing popularity in machine learning recently, with applications ranging from meta-learning [20], to hyperparameter optimization [45, 21] and neural architecture search [37]. We use the framework of bilevel optimization to generate coresets. Closest to our work, bilevel optimization was used in [47] to reweigh samples to overcome training set imbalance or corruption, and sensor selection was approached via bilevel optimization in [51]. We note that, while the latter work uses a similar strategy for sensor subset selection to ours, we investigate the different setting of weighted data summarization for continual learning and streaming with neural networks.

## 3 Coresets via Bilevel Optimization

We first present our coreset construction given a *fixed* dataset $\mathcal{X} = \{(x_i, y_i)\}_{i=1}^n$.[3] We consider a weighted variant of empirical risk minimization (ERM) where our goal is to minimize $L(\theta, w) = \sum_{i=1}^n w_i \ell_i(\theta)$, where $\ell_i(\theta) = \ell(x_i, y_i; \theta)$, and $w = \{w_1, \ldots, w_n\} \in \mathbb{R}_+^n$ is a set of positive weights. Standard ERM is recovered when $w_i = 1$, $\forall i \in [n]$, in which case we simply write $L(\theta)$. A *coreset* is a weighted subset of $\mathcal{X}$, equivalently represented by a sparse vector $\hat{w}$, where points having zero weights are not considered to be part of the coreset. A good coreset ensures that $L(\theta, \hat{w})$ is a good proxy for $L(\theta)$, i.e., optimizing on $L(\theta, \hat{w})$ over $\theta$ yields good solutions when evaluated on $L(\theta)$. In this spirit, a natural goal thus would be to determine a set of weights $\hat{w}$, such that we minimize

$$\hat{w} \in \underset{w \in \mathbb{R}_+^n, \|w\|_0 \leq m}{\arg\min} L(\theta^*(w)) \text{ s.t. } \theta^*(w) \in \underset{\theta}{\arg\min} L(\theta, w), \tag{1}$$

where $m \leq n$ is a constraint on the coreset size. This problem is an instance of *bilevel optimization*, where we minimize an *outer* objective, here $L(\theta^*(w))$, which in turn depends on the solution $\theta^*(w)$ to an *inner* optimization problem $\arg\min_\theta L(\theta, w)$. Before presenting our algorithm for solving (1), we present some background on bilevel optimization.

## 3.1 Background: Bilevel Optimization

Suppose $g : \Theta \times \mathcal{D} \to \mathbb{R}$ and $f : \Theta \times \mathcal{D} \to \mathbb{R}$, then the general form of bilevel optimization is

$$
\begin{aligned}
\min_{w \in \mathcal{D}} \quad & G(w) := g(\theta^*(w), w) \\
\text{s.t.} \quad & \theta^*(w) \in \arg\min_{\theta \in \Theta} f(\theta, w).
\end{aligned}
\tag{2}
$$

Our data summarization problem is recovered with $g(\theta^*(w), w) = L(\theta^*(w))$ and $f(\theta, w) = L(\theta, w)$. The general optimization problem above is known to be NP-hard even if both sets $\Theta$ and $\mathcal{D}$ are polytopes and both $g$ and $f$ are linear. Provably correct solvers rely on branch and bound techniques [5]. For differentiable $g$ and twice differentiable $f$, the above problem can however be heuristically solved via first-order methods. Namely, the constraint $\theta^*(w) \in \arg\min_{\theta \in \Theta} f(\theta, w)$ can be relaxed to $\frac{\partial f(\theta^*, w)}{\partial \theta^*} = 0$. This relaxation is tight if $f$ is strictly convex. A crucial result that allows us apply first-order methods by enabling the calculation of the gradient of $G$ with respect to $w$ is the *implicit function theorem* applied to $\frac{\partial f(\theta^*, w)}{\partial \theta^*} = 0$. Combined with the the total derivative and the chain rule,

$$
\frac{dG(w)}{dw} = \frac{\partial g}{\partial w} - \frac{\partial g}{\partial \theta} \cdot \left( \frac{\partial^2 f}{\partial \theta \partial \theta^\top} \right)^{-1} \cdot \frac{\partial^2 f}{\partial w \partial \theta^\top},
\tag{3}
$$

where the terms are evaluated at $\theta = \theta^*(w)$. At each step of the gradient descent on the outer objective, the inner objective needs to be solved to optimality. While this heuristic may be converging only to a stationary point [24], such solutions are successful in many applications [45, 51].

## 3.2 Warm-up: Least Squares Regression, and Connections to Experimental Design

We first instantiate our approach to the problem of weighted and regularized least squares regression. In this case, the inner optimization problem, $\hat{\theta}(w) = \arg\min_\theta \sum_{i=1}^n w_i(x_i^\top \theta - y_i)^2 + \lambda \|\theta\|_2^2$, admits a closed-form solution. For this special case there are natural connections to the literature on *optimal experimental design*, a well-studied topic in statistics [8].

The data summarization problem with outer objective $g(\hat{\theta}) = \sum_{i=1}^n (x_i^\top \hat{\theta}(w) - y_i)^2$ is closely related to *Bayesian V-optimal design*. Namely, under standard Bayesian linear regression assumptions $y = X\theta + \epsilon$, $\epsilon \sim \mathcal{N}(0, \sigma^2 I)$ and $\theta \sim \mathcal{N}(0, \lambda^{-1})$, the summarization objective $\mathbb{E}_{\epsilon, \theta}[g(\hat{\theta})]$ and the Bayesian V-experimental design outer objective $g_V(\hat{\theta}) = \mathbb{E}_{\epsilon, \theta}[\|X^\top(\hat{\theta} - \theta)\|_2^2]$ differ by $\sigma^2/2$ in the infinite data limit, as shown in Proposition 6 in Appendix A. Consequently, it can be argued that, in the large data limit, the optimal coreset with binary weights corresponds to the solution of Bayesian V-experimental design. We defer the detailed discussion to Appendix A.

Using $g_V$ as our outer objective, solving the inner objective in closed form, we identify the Bayesian V-experimental design objective,

$$
G(w) = \frac{1}{2n} \mathbf{Tr} \left( X \left( \frac{1}{\sigma^2} X^\top D(w) X + \lambda I \right)^{-1} X^\top \right),
$$

where $D(w) := \mathrm{diag}(w)$. In Lemma 8 in Appendix A we show that $G(w)$ is smooth and convex in $w$ when the integrality is relaxed. This, together with the relationship between $g$ and $g_V$, suggests that first order methods for solving the bilevel data summarization objective can be expected to succeed, at least in the large data limit and the special case of least squares regression.

## 3.3 Incremental Subset Selection

One challenge that we have not addressed yet is how to deal with the cardinality constraint $\|w\|_0 \leq m$. A natural attempt is, in the spirit of the Lasso [52], to transform the $\|w\|_0$ constraint into the $\|w\|_1$ constraint. However, the solution of the inner optimization is unchanged if the weights

and the regularizer are rescaled by a common factor, rendering norm-based sparsity regularization meaningless. A similar observation was made in [51]. On the other hand, a greedy combinatorial approach would treat $G$ as a set function and increment the set of selected points by inspecting marginal gains. It turns out that Bayesian V-experimental design is approximately *submodular*, as shown in Proposition 7 in Appendix A. As a consequence, at least for the linear regression case, such greedy algorithms produce provably good solutions, if we restrict the weights to be binary [15, 26]. Unfortunately, for general losses the greedy approach comes at a significant cost: at each step, the bilevel optimization problem of finding the optimal weights must be solved for each point which may be added to the coreset. This makes greedy selection impractical.

**Selection using Matching Pursuit.** In this work, we propose an efficient solution summarized in Algorithm 1 based on *cone constrained generalized matching pursuit* [38]. With the atom set $\mathcal{A}$ corresponding to the standard basis of $\mathbb{R}^n$, generalized matching pursuit proceeds by incrementally increasing the active set of atoms that represents the solution by selecting the atom that minimizes the linearization of the objective at the current iterate. The benefit of this approach is that incrementally growing the atom set can be stopped when the desired size $m$ is reached and thus the $\|w\|_0 \leq m$ constraint is active. We use the

---

**Algorithm 1** Coresets via Bilevel Optimization

**Input:** Data $\mathcal{X} = \{(x_i, y_i)\}_{i=1}^n$, coreset size $m$
**Output:** weights $w$ encoding the coreset
$w = [0, \ldots, 0]$
Choose $i \in [n]$ randomly, set $w_i = 1$, $S_1 = \{i\}$.
**for** $t \in [2, ..., m]$ **do**
  Find $w^*_{S_{t-1}}$ local min of $G(w)$ by projected GD
  s.t. $\text{supp}(w^*_{S_{t-1}}) = S_{t-1}$ .
  $k^* = \arg\min_{k \in [n]} \nabla_{w_k} G(w^*_{S_{t-1}})$
  $S_t = S_{t-1} \cup \{k^*\}$, $w_{k^*} = 1$
**end for**

---

fully-corrective variant of the algorithm, where, once a new atom is added, the weights are fully reoptimized by gradient descent using the implicit gradient (Eq. (3)) with projection to positive weights.

Suppose a set of atoms $S_t \subset \mathcal{A}$ of size $t$ has already been selected. Our method proceeds in two steps. First, the bilevel optimization problem (1) is restricted to weights $w$ having support $S_t$. Then we optimize to find the weights $w^*_{S_t}$ with domain support restricted to $S_t$ that represent a local minimum of $G(w)$ defined in Eq. 2 with $g(\theta^*(w), w) = L(\theta^*(w))$ and $f(\theta, w) = L(\theta, w)$. Once these weights are found, the algorithm increments $S_t$ with the atom that minimizes the first order Taylor expansion of the outer objective around $w^*_{S_t}$,

$$k^* = \arg\min_{k \in [n]} e_k^\top \nabla_w G(w^*_{S_t}), \tag{4}$$

where $e_k$ denotes the $k$-th standard basis vector of $\mathbb{R}^n$. In other words, the chosen point is the one with the largest negative implicit gradient (Eq. (3)).

We can gain an insights into the selection rule in Eq. (4) by expanding $\nabla_w G$ using Eq. (3). For this, we use the inner objective $f(\theta, w) = \sum_{i=1}^n w_i \ell_i(\theta)$ without regularization for simplicity. Noting that $\frac{\partial^2 f(\theta, w)}{\partial w_k \partial \theta^\top} = \nabla_\theta \ell_k(\theta)$ we can expand Eq. (4) to get,

$$k^* = \arg\max_{k \in [n]} \nabla_\theta \ell_k(\theta)^\top \left( \frac{\partial^2 f(\theta, w^*_{S_t})}{\partial \theta \partial \theta^\top} \right)^{-1} \nabla_\theta g(\theta), \tag{5}$$

where the gradients and partial derivatives are evaluated at $w^*_{S_t}$ and $\theta^*(w^*_{S_t})$. Thus with the choice $g(\theta) = \sum_{i=1}^n \ell_i(\theta)$, the selected point's gradient has the largest bilinear similarity with $\nabla_\theta \sum_{i=1}^n \ell_i(\theta)$, where the similarity is parameterized by the inverse Hessian of the inner problem.

**Theoretical Guarantees.** If the overall optimization problem is convex (as in the case of Bayesian V-experimental design, Lemma 8 in Appendix A), one can show that cone constrained generalized matching pursuit provably converges. Following [38], the convergence of the algorithm depends on properties of the atom set $\mathcal{A}$, which we do not review here in full.

**Theorem 1** (cf. Theorem 2 of [38]). *Let $G$ be $L$-smooth and convex. After $t$ iterations in Algorithm 1 we have,*

$$G(w^*_{S_t}) - G^* \leq \frac{8L + 4\epsilon_1}{t + 3},$$

*where $t \leq m$ (number of atoms), and $\epsilon_1 = G(w^*_{S_1}) - G^*$ is the suboptimality gap at $t = 1$.*

Thus, by iterating $m - 1$ times, we reach the cardinality constraint. Note that by imposing a bound on the weights as commonly done in experimental design [18], our algorithm would be equivalent to Frank-Wolfe [22]. While in general the function $G$ might not be convex for more complex models, we nevertheless demonstrate the effectiveness of our method empirically in such scenarios in Section 6.

### 3.4 Relation to Influence Functions

It turns out our approach is also related to incremental subset selection via *influence functions*. The empirical influence function, known from robust statistics [13], denotes the effect of a single sample on the estimator. Influence functions have been recently used in [34] to understand the dependence of neural network predictions on a single training point and to generate adversarial training examples. To uncover the relationship of our method to influence functions, let us consider the influence of the $k$-th point on the outer objective. Suppose that we have already selected the subset $S$ and found the corresponding weights $w_S^*$. Then, the influence of point $k$ on the outer objective is

$$\mathcal{I}(k) := -\left.\frac{\partial \sum_{i=1}^n \ell_i(\theta^*)}{\partial \varepsilon}\right|_{\varepsilon=0}, \quad \text{s.t.} \quad \theta^* = \arg\min_\theta \sum_{i=1}^n w_{S,i}^* \ell_i(\theta) + \varepsilon \boldsymbol{\ell_k}(\boldsymbol{\theta}).$$

Following [34] and using the result of [14], under twice differentiability and strict convexity of the inner loss, the empirical influence function at $k$ is $\left.\frac{\partial \theta^*}{\partial \varepsilon}\right|_{\varepsilon=0} = -\left(\frac{\partial^2 \sum_{i=1}^n w_{S,i}^* \ell_i(\theta^*)}{\partial\theta\partial\theta^\top}\right)^{-1} \nabla_\theta \ell_k(\theta^*)$. Now, applying the chain rule to $\mathcal{I}(k)$, as shown in Proposition 9 in Appendix B, we can see that $\arg\max_k \mathcal{I}(k)$ and the selection rule in Equation (5) are the same.

## 4 Coresets for Neural Networks

While our proposed coreset framework is generally applicable to any twice differentiable model, we showcase it for the challenging case of deep neural networks. In applying Algorithm 1 to a neural network with a large number of parameters, inverting the Hessian in each outer iteration (Eq. (3)) is an impeding factor. Several works propose Hessian-vector product approximations, for example, through the conjugate gradient algorithm [45] or Neumann series [40]. While applicable in our setting, these methods become costly depending on the number of coreset points and outer iterations.

We propose an alternative approach that results in a large speedup for small coreset size (max. 500 points). The key idea is to use a *proxy model* in the bilevel optimization that provides a good data summary for the original model. In this work, our choice for proxies are functions in a reproducing kernel Hilbert space (RKHS) $\mathcal{H}$ with associated kernel $k$, possibly adjusted to the neural network architecture. We assume $k$ to be positive definite for simplicity and we use the same convex loss as for the neural network. Now, the coreset generation problem with regularized inner objective transforms into

$$\min_{w\in\mathbb{R}_+^n,\, \|w\|_0\leq m} g(h^*) \text{ s.t. } h^* = \arg\min_{h\in\mathcal{H}} \sum_{i=1}^n w_i \ell_i(h) + \lambda \|h\|_{\mathcal{H}}^2.$$

The regularizer $\lambda$ could also be optimized jointly with $w$, but we use fixed values in our applications. Now suppose we have selected a subset of atoms and denote their index set as $S$. From the *representer theorem* we know that the inner problem admits the representation $h^*(\cdot) = \alpha^\top K_{S,.}$, where $\alpha \in \mathbb{R}^{|S|}$ and $K$ is the Gram matrix associated with the data. Now the bilevel optimization problem for optimizing the weights takes the form

$$\min_{w\in\mathbb{R}_+^n,\, \text{supp}(w)=S} g(\alpha^{*\top} K_{S,.}), \quad \text{s.t. } \alpha^* = \arg\min_{\alpha\in\mathbb{R}^{|S|}} \sum_{i\in S} w_i \ell_i(\alpha^\top K_{S,i}) + \lambda\alpha^\top K_{S,S}\alpha. \quad (6)$$

That is, with the help of the representer theorem, we can reduce the size of the inner level parameters to at most the size $m$ of the coreset. This allows us to use fast solvers for the inner problem, and to use the aforementioned approximate inverse Hessian-vector product methods when calculating the implicit gradient. In this work, we use the conjugate gradient method [45].

The choice of $\mathcal{H}$ is crucial for the success of the proxy reformulation. For neural networks, we propose to use as proxies their corresponding *Neural Tangent Kernels (NTK)* [30], which are fixed kernels characterizing the network's training with gradient descent in the infinite-width limit. While other proxy choices are possible (e.g., fixed last-layer embedding with linear kernel on top, or even RBF kernel, see Appendix D) we leave their investigation for future work.

**Computational Cost.** In the standard formulation, each inner iteration is performed using a small number of SGD steps to find an approximate minimizer of the inner optimization problem. The bottleneck is introduced by the outer descent step: the implicit gradient approximation via 30 conjugate gradient steps requires one minute for a ResNet-18 on a GPU, which makes the approach impractical for weighted coreset selection. The proxy reformulation reduces the number of parameters to $\mathcal{O}(m)$ and its time complexity depends cubically on the coreset size $m$. As a result, an outer descent step is $200\times$ faster in the proxy compared to the standard formulation. On the other hand, the proxy reformulation introduces the overhead of calculating the proxy kernel — for calculating the NTK efficiently, we rely on the library of [44]. We measure the runtime of coreset generation in the proxy formulation in Section 6.3. Further speedups are discussed in Appendix E.

**Limitations.** While our proposed framework excels at small coreset sizes, generating summaries larger than 500 incurs significant computational overhead. A possible remedy is to perform the greedy selection step in batches or to perform greedy elimination instead of forward selection. In terms of theoretical guarantees, due to the hardness of the cardinality-constrained bilevel optimization problem, our method is a heuristic for coreset selection for neural networks.

## 5    Applications in Continual Learning and Streaming Deep Learning

We now demonstrate how our coreset construction can achieve significant performance gains in continual learning and in the more challenging streaming settings with neural networks. We build on approaches that alleviate catastrophic forgetting by keeping representative past samples in a replay memory. Our goal is to compare our method to other data summarization strategies for managing the replay memory. We keep the network structure fixed during training. This is termed as the "single-head" setup, which is more challenging than instantiating new top layers for different tasks ("multi-head" setup) and does not assume any knowledge of the task descriptor during training and test time [17].

For *continual learning with replay memory* we employ the following protocol. The learning algorithm receives data $\mathcal{X}_1, \ldots, \mathcal{X}_T$ arriving in order from $T$ different tasks. At time $t$, the learner receives $\mathcal{X}_t$ but can only access past data through a small number of samples from the replay memory of size $m$. We assume that equal memory is allocated for each task in the buffer, and that the summaries $\mathcal{C}_1, \ldots, \mathcal{C}_T$ are created per task, with weights equal to 1. Thus, the optimization objective at time $t$ is

$$\min_{\theta} \frac{1}{|\mathcal{X}_t|} \sum_{(x,y)\in\mathcal{X}_t} \ell(x,y;\theta) + \beta \sum_{\tau=1}^{t-1} \frac{1}{|\mathcal{C}_\tau|} \sum_{(x,y)\in\mathcal{C}_\tau} \ell(x,y;\theta),$$

where $\sum_{\tau=1}^{t-1} |\mathcal{C}_\tau| = m$, and $\beta$ is a hyperparameter controlling the regularization strength of the loss on the samples from the replay memory. After performing the optimization, $\mathcal{X}_t$ is summarized into $\mathcal{C}_t$ and added to the buffer, while previous summaries $\mathcal{C}_1, \ldots, \mathcal{C}_{t-1}$ are shrunk such that $|\mathcal{C}_\tau| = \lfloor m/t \rfloor$. The shrinkage is performed by running the summarization algorithms on each $\mathcal{C}_1, \ldots, \mathcal{C}_{t-1}$ again, which for greedy strategies is equivalent to retaining the first $\lfloor m/t \rfloor$ samples from each summary.

The *streaming setting* is more challenging: we assume the learner is faced with small data batches $\mathcal{X}_1, ..., \mathcal{X}_T$ arriving in order, but the batches do not contain information about the task boundaries. In fact, even the notion of tasks might not be defined. Denoting by $\mathcal{M}_t$ the replay memory at time $t$, the optimization objective at time $t$ for learning under streaming with replay memory is

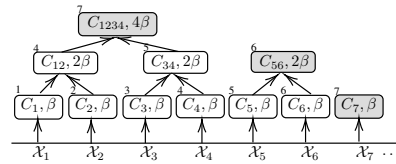

$$\min_{\theta} \frac{1}{|\mathcal{X}_t|} \sum_{(x,y)\in\mathcal{X}_t} \ell(x,y;\theta) + \frac{\beta}{|\mathcal{M}_{t-1}|} \sum_{(x,y)\in\mathcal{M}_{t-1}} \ell(x,y;\theta).$$

Figure 2: Merge-reduce on 7 steps with a buffer with 3 slots. The grey nodes are in the buffer after the 7 steps, the numbers in the upper left corners represent the construction time of the corresponding coresets.

**Streaming Coresets via Merge-Reduce.** Managing the replay memory is crucial for the success of our method in streaming. We offer a principled way to achieve this, naturally supported by our framework, using the following idea: two coresets can be summarized into a single one by applying our bilevel construction with the outer objective as the loss on the union of the two coresets. Relying on this idea, we use a variant of the merge-reduce framework of [10]. For this,

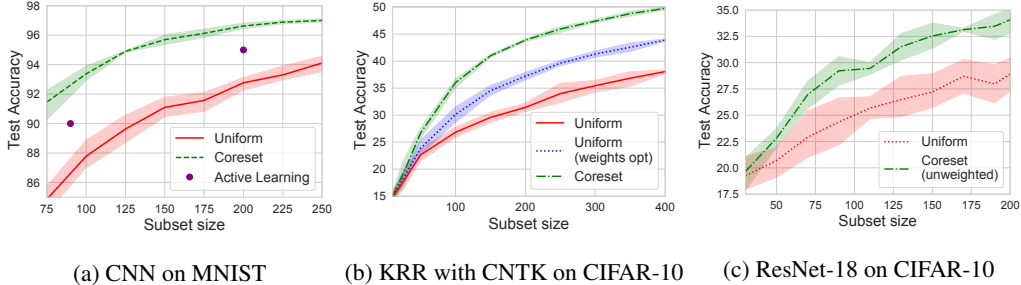

| (a) CNN on MNIST | (b) KRR with CNTK on CIFAR-10 | (c) ResNet-18 on CIFAR-10 |

Figure 3: a) Performance of a CNN trained on subsets of MNIST. b) Kernel ridge regression (KRR) on subsets of CIFAR-10. Our method obtains almost 50% test accuracy on CIFAR-10 when trained on only 400 points. c) ResNet-18 trained on subsets of CIFAR-10. Coresets have binary weights.

we divide the buffer into $s$ equally-sized slots. We associate regularizers $\beta_i$ with each of the slots, which will be *proportional to the number of points* they represent. A new batch is compressed into a new slot with associated $\beta$ and it is appended to the buffer, which now might contain an extra slot. The reduction to size $m$ happens as follows: select consecutive slots $i$ and $i+1$ based on Algorithm 3 in Appendix C, then join the contents of the slots (*merge*) and create the coreset of the merged data (*reduce*). The new coreset replaces the two original slots with $\beta_i + \beta_{i+1}$ associated with it (Algorithm 2 in Appendix C). We illustrate the merge-reduce coreset construction for a buffer with 3 slots and 7 steps in Figure 2.

# 6 Experiments

We demonstrate the empirical effectiveness of our method in various settings. In all our experiments we generate the coreset via our proxy formulation. For each neural network architecture we calculate the corresponding (convolutional) neural tangent kernel without pooling using the library of [44].

## 6.1 Dataset summarization

We showcase our method by training a convolutional neural network (CNN) on a small subset of MNIST selected by our coreset construction. The CNN consists of two blocks of convolution, dropout, max-pooling and ReLU activation, where the number of filters are 32 and 64 and have size 5x5, followed by two fully connected layers of size 128 and 10 with dropout. The dropout probability is 0.5. The CNN is trained on the data summary using Adam with learning rate $5 \cdot 10^{-4}$. The results for summarizing MNIST are shown in Figure 3a, where we plot the test accuracy against the summary size over 5 random seeds for uniform sampling, coreset generation, and for the state-of-the-art in active learning for our chosen CNN architecture [33].

Our next experiment follows [3] and solves classification on CIFAR-10 [35] via kernelized ridge regression (KRR) applied to the one-hot-encoded labels $Y$. KRR can be solved on a subset $S$ in closed-form $\alpha^* = (D(w_S)K_{S,S} + \lambda\mathbb{I})^{-1}D(w_S)Y_S$, which replaces the inner optimization problem. We solve the coreset selection for KRR using the CNTK proposed in [3] with 6 layers and global average pooling, with normalization. We also experiment with uniform sampling of points with weights optimized through bilevel optimization. The results are shown in Figure 3b. We observe that with CNTK we can obtain a test accuracy of almost 50% with only 400 samples. We further validate our proxy formulation by selecting subsets of CIFAR-10 and training a variant of ResNet-18 [28] without batch normalization on the chosen subsets. We restrict the coreset weights to binary in order to show that choosing representative *unweighted* points can alone improve the performance, as illustrated in Figure 3c.

## 6.2 Continual Learning

We next validate our method in the replay memory-based approach to continual learning. We use the following 10-class classification datasets:

- **PermMNIST** [25]: consist of 10 tasks, where in each task all images' pixels undergo the same fixed random permutation.

Table 1: Continual learning with replay memory size of 100 for versions of MNIST and 200 for CIFAR-10. We report the average test accuracy over the tasks over 5 runs with different random seeds. Our coreset construction performs among the best on all datasets.

| Method | PermMNIST | SplitMNIST | SplitCIFAR-10 |
|---|---|---|---|
| Uniform sampling | $78.46 \pm 0.40$ | $92.80 \pm 0.79$ | $\mathbf{36.20 \pm 3.19}$ |
| $k$-means of features | $78.34 \pm 0.49$ | $93.40 \pm 0.56$ | $33.41 \pm 2.48$ |
| $k$-center of embeddings | $78.57 \pm 0.58$ | $93.84 \pm 0.78$ | $\mathbf{36.91 \pm 2.42}$ |
| Hardest samples | $76.79 \pm 0.55$ | $89.62 \pm 1.23$ | $28.10 \pm 1.79$ |
| iCaRL's selection | $\mathbf{79.68 \pm 0.41}$ | $93.99 \pm 0.39$ | $34.52 \pm 1.62$ |
| **Coreset** | $\mathbf{79.26 \pm 0.43}$ | $\mathbf{95.87 \pm 0.20}$ | $\mathbf{37.60 \pm 2.41}$ |

Table 2: Upper: VCL with 20 summary points / task. VCL can benefit from our coreset. Lower: Streaming with buffer size 100. Coreset methods use the merge-reduce buffer.

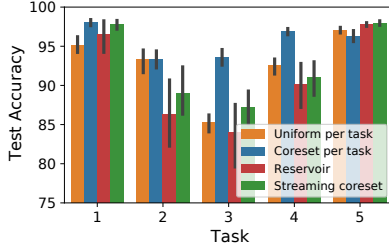

| | Method | PermMNIST | SplitMNIST |
|---|---|---|---|
| **VCL** | $k$-center | $85.33 \pm 0.67$ | $65.71 \pm 3.17$ |
| | Uniform | $84.96 \pm 0.17$ | $80.06 \pm 2.19$ |
| | **Coreset** | $\mathbf{86.18 \pm 0.21}$ | $\mathbf{84.66 \pm 0.66}$ |
| **Stream** | Train on coreset only | $45.03 \pm 1.31$ | $89.99 \pm 0.76$ |
| | Reservoir sampling | $73.21 \pm 0.59$ | $90.72 \pm 0.97$ |
| | **Streaming coreset** | $\mathbf{74.44 \pm 0.52}$ | $\mathbf{92.59 \pm 1.20}$ |

Figure 4: Per-task test accuracy on SplitMNIST. Out method represents most of the tasks better than uniform / reservoir sampling.

- **SplitMNIST** [57]: MNIST is split into 5 tasks, where each task consists of distinguishing between consecutive image classes.
- **SplitCIFAR-10**: similar to SplitMNIST on CIFAR-10.

We keep a subsample of 1000 points for each task for all datasets, while we retain the full test sets. For PermMNIST we use a fully connected net with two hidden layers with 100 units, ReLU activations, and dropout with probability 0.2 on the hidden layers. For SplitMNIST we use the CNN described in Section 6.1. We fix the replay memory size $m = 100$ for these task. For SplitCIFAR-10 we the ResNet presented in Section 6.1 and set the memory size to $m = 200$. We train our networks for 400 epochs using Adam with step size $5 \cdot 10^{-4}$ after each task.

We perform an exhaustive comparison of our method to other data selection methods proposed in the continual learning or the coreset literature, under the protocol described in Section 5. These include, among others, $k$-center clustering in last layer embedding [49] and feature space [43], iCaRL's selection [46] and retaining the hardest-to-classify points [1]. For each method, we report the test accuracy averaged over tasks on the best buffer regularization strength $\beta$. For a fair comparison to other methods in terms of summary generation time, we restrict our method in all of the continual learning and streaming experiments to using *binary coreset weights* only. A selection of results is presented in Table 1, while the full list is available in Appendix D. Our coreset construction consistently performs among the best on all datasets. In Appendix D we present a study of the effect of the replay memory size.

Our method can also be combined with different approaches to continual learning, such as VCL [43]. While VCL also relies on coresets, it was proposed with uniform and $k$-center summaries. We replace these with our coreset construction, and, following [43], we conduct an experiment using a single-headed two-layer network with 256 units per layer and ReLU activations, where the coreset size is set to 20 points per task. The results in Table 2 corroborate the advantage of our method over simple selection rules, and suggest that VCL can benefit from representative coresets.

## 6.3 Streaming

We now turn to the more challenging streaming setting, which is oblivious to the existence of tasks. For this experiment, we modify PermMNIST and SplitMNIST by first concatenating all tasks for each dataset and then streaming them in batches of size 125. We fix the replay memory size to $m = 100$

and set the number of slots $s = 10$. We train our networks for 40 gradient descent steps using Adam with step size $5 \cdot 10^{-4}$ after each batch. We use the same architectures as in the previous experiments.

We compare our coreset selection to reservoir sampling [55] and the sample selection methods of [2] and [27]. We were unable to tune the latter two to outperform reservoir sampling, except [2] on PermMNIST, achieving test accuracy of $74.43 \pm 1.02$. Table 2 confirms the dominance of our strategy over the competing methods and the validity of the merge-reduce framework. For inspecting the gains obtained by our method over uniform / reservoir sampling, we plot the final per task test accuracy on SplitMNIST in Figure 4. We notice that the advantage of the coreset method does not come from excelling at one particular task, but rather by representing the majority of tasks better than uniform sampling. We have also experimented with streaming on CIFAR-10 with buffer size $m = 200$, where our coreset construction did not outperform reservoir sampling. However, when the task representation in the stream is imbalanced, our method has significant advantages, as we show in the following experiment.

Table 3: Imbalanced streaming on SplitMNIST and SplitCIFAR-10. Our proposed method is competitive with strategies designed for imbalanced streams.

| Method | SplitMNIST | SplitCIFAR-10 |
|---|---|---|
| Reservoir | $80.60 \pm 4.36$ | $27.22 \pm 1.24$ |
| CBRS | $89.71 \pm 1.31$ | $\mathbf{32.25 \pm 1.69}$ |
| **Coreset** | $\mathbf{92.30 \pm 0.23}$ | $\mathbf{33.98 \pm 1.44}$ |

Table 4: Runtimes for generating coresets out of 1000 points with CNTKs for the CNN and ResNet-18 described in Sec. 6.1.

| Op / CNTK | CNN | ResNet-18 |
|---|---|---|
| **Kernel calc.** | 6.3 s | 56.2 s |
| **Coreset 100** | 21.2 s | 25.4 s |
| **Coreset 400** | 186.7 s | 188.9 s |

**Imbalanced Streaming.** The setup of the streaming experiment favors reservoir sampling, as the data in the stream from different tasks is balanced. We illustrate the benefit of our method in the more challenging scenario when the task representation is *imbalanced*. Similarly to [2], we create imbalanced streams from SplitMNIST and SplitCIFAR-10, by retaining 200 random samples from the first four tasks and 2000 from the last task. In this setup, reservoir sampling will underrepresent the first tasks. For SplitMNIST we set the replay buffer size to $m = 100$ while for SplitCIFAR-10 we use $m = 200$. We evaluate the test accuracy on the tasks individually, where we do not undersample the test set. We train on the two imbalanced streams the CNN and the ResNet-18 described in Section 6.1, and set the number of slots to $s = 1$. We compare our method to reservoir sampling and class-balancing reservoir sampling (CBRS) [11]. The results in Table 3 confirm the flexibility of our framework which is competitive with methods specifically designed for imbalanced streams.

**Runtime.** In Table 4 we measure the runtime of selecting 100 and 400 coreset points from a batch of 1000 points with our proxy formulation, with the CNTK corresponding to the two convolutional networks used in the experiments. The CNTKs are calculated on a GeForce GTX 1080 Ti GPU while the coreset selection is performed on a single CPU. Due to the non-linear increase of computation time in terms of the coreset size, our proxy reformulation is most practical for smaller coreset sizes.

## 7 Conclusion

We presented a novel framework for coreset generation based on bilevel optimization with cardinality constraints. We theoretically established connections to experimental design and empirical influence functions. We showed that our method yields representative data summaries for neural networks and illustrated its advantages in alleviating catastrophic forgetting in continual learning and streaming deep learning, where our coreset construction performs among the best summarization strategies.

## Broader Impact

Coresets can efficiently handle large datasets under computational constraints, thus significantly reducing computational costs and possibly the energy consumption for applications. They also provide an avenue towards limiting the amount of data that needs to be stored, with possible privacy benefits. Explicitly optimizing representativeness of the retained samples beyond accuracy (e.g., for counteracting biases in the data) is a promising direction for future work.

## Acknowledgments and Disclosure of Funding

This research was supported by the SNSF grant 407540_167212 through the NRP 75 Big Data program and by the European Research Council (ERC) under the European Union's Horizon 2020 research and innovation programme grant agreement No 815943.

## Footnotes

[1] Our coreset library is available at `https://github.com/zalanborsos/bilevel_coresets`.

[2]Although these methods were not designed for replay-based continual learning or streaming, they can be employed in these settings; we thus compare them with our method empirically.

[3]Our construction also applies in the unsupervised setting.

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
