[Supplementary Material]

# Supplementary Material

## A    Connections to Experimental Design

In this section, the weights are assumed to be binary, i.e., $w \in \{0, 1\}^n$. We will use a shorthand $X_S$ for matrix where only rows of X whose indices are in $S \subset [n]$ are selected. This will be equivalent to selection done via diagonal matrix $D(w)$, where $i \in S$ corresponds to $w_i = 1$ and zero otherwise.

Additionally, let $\hat{\theta}$ be a minimizer of the following loss,

$$\hat{\theta} = \arg\min_{\theta} \sum_{i=1}^{n} w_i (x_i^\top \theta - y_i)^2 + \lambda \sigma^2 ||\theta||_2^2 \tag{7}$$

which has the following closed form,

$$\hat{\theta}_S = (X_S^\top X_S + \lambda \sigma^2 I)^{-1} X_S^\top y_S. \tag{8}$$

**Frequentist Experimental Design.**    Under the assumption that the data follows a linear model $y = X\theta + \epsilon$, where $\epsilon \sim \mathcal{N}(0, \sigma^2)$, we can show that the bilevel coreset framework instantiated with the inner objective (7) and $\lambda = 0$, with various choices of outer objectives is related to frequentist optimal experimental design problems. The following propositions show how different outer objectives give rise to different experimental design objectives.

**Proposition 2** (A-experimental design). *Under the linear regression assumptions and when $g(\hat{\theta}) = \frac{1}{2}\mathbb{E}_\epsilon \left[\left\|\theta - \hat{\theta}\right\|_2^2\right]$, with the inner objective is equal to (7) with $\lambda = 0$, the objective simplifies,*

$$G(w) = \frac{\sigma^2}{2}\mathbf{Tr}((X^\top D(w)X)^{-1}).$$

*Proof.* Using the closed from in (8), and model assumptions, we see that $\hat{\theta} = \theta + (X_S^\top X_S)^{-1} X_S^\top \epsilon_S$. Plugging this in to the outer objective,

$$g(\hat{\theta}) = \frac{1}{2}\mathbb{E}_\epsilon \left[\left\|\theta - \hat{\theta}\right\|_2^2\right] \tag{9}$$

$$= \frac{1}{2}\mathbb{E}_\epsilon \left[\left\|(X_S^\top X_S)^{-1} X_S^\top \epsilon_S\right\|_2^2\right] \tag{10}$$

$$= \frac{1}{2}\mathbb{E}_\epsilon \left[\mathbf{Tr}\left(\epsilon_S^\top X_S (X_S^\top X_S)^{-2} X_S^\top \epsilon_S\right)\right] \tag{11}$$

$$= \frac{\sigma^2}{2}\mathbf{Tr}\left((X_S^\top X_S)^{-1}\right) \tag{12}$$

$$= \frac{\sigma^2}{2}\mathbf{Tr}\left((X^\top D(w)X)^{-1}\right) \tag{13}$$

where in the third line we used the cyclic property of trace and subsequently the normality of $\epsilon$.    $\square$

**Proposition 3** (V-experimental design). *Under the linear regression assumptions and when $g(\hat{\theta}) = \frac{1}{2n}\mathbb{E}_\epsilon \left[\left\|X\theta - X\hat{\theta}\right\|_2^2\right]$ and the inner objective is equal to (7) with $\lambda = 0$, the objective simplifies,*

$$G(w) = \frac{\sigma^2}{2n}\mathbf{Tr}(X(X^\top D(w)X)^{-1}X^\top).$$

*Proof.* Using the closed from in (8), and model assumptions, we see that $\hat{\theta} = \theta + (X_S^\top X_S)^{-1} X_S^\top \epsilon_S$. Plugging this in to the outer objective $g(\hat{\theta})$,

$$G(w) = \frac{1}{2n} \mathbb{E}_\epsilon \left[ \left\| X\theta - X\hat{\theta} \right\|_2^2 \right] \tag{14}$$

$$= \frac{1}{2n} \mathbb{E}_\epsilon \left[ \left\| X(X_S^\top X_S)^{-1} X_S^\top \epsilon_S \right\|_2^2 \right] \tag{15}$$

$$= \frac{1}{2n} \mathbb{E}_\epsilon \left[ \mathbf{Tr} \left( \epsilon_S^\top X_S (X_S^\top X_S)^{-1} X^\top X (X_S^\top X_S)^{-1} X_S^\top \epsilon_S \right) \right] \tag{16}$$

$$= \frac{\sigma^2}{2n} \mathbf{Tr} \left( X(X_S^\top X_S)^{-1} X^\top \right) \tag{17}$$

$$= \frac{\sigma^2}{2n} \mathbf{Tr} \left( X(X^\top D(w) X)^{-1} X^\top \right) \tag{18}$$

where in the third line we used the cyclic property of trace and subsequently the normality of $\epsilon$. $\square$

**Infinite data limit.** The following proposition links the data summarization objective and V-experimental design in infinite data limit $n \to \infty$.

**Proposition 4** (Infinite data limit). *Under the linear regression assumptions, let $g_V$ be*

$$g_V(\hat{\theta}) = \frac{1}{2n} \mathbb{E}_\epsilon \left[ \left\| X\theta - X\hat{\theta} \right\|_2^2 \right]$$

*the V-experimental design outer objective, and let the summarization objective be,*

$$g(\hat{\theta}) = \frac{1}{2n} \mathbb{E}_\epsilon \left[ \sum_{i=1}^n (x_i^\top \hat{\theta} - y_i)^2 \right].$$

*Let $\Theta$ be the set containing random variables $\hat{\theta}$ representing the inner problem's solution on a finite subset of the data. Thus each $\hat{\theta}$ depends on a finite set of $\epsilon_S := \cup_{i \in S}\{e_i\}$, where $S \subset [n]$. Then*

$$\lim_{n \to \infty} g(\hat{\theta}) - g_V(\hat{\theta}) - \frac{\sigma^2}{2} = 0, \quad \forall \hat{\theta} \in \Theta.$$

*Proof.* Since $y_i = x_i^\top \theta + \epsilon_i$, we have,

$$g(\hat{\theta}) = \frac{1}{2n} \mathbb{E}_\epsilon \left[ \sum_{i=1}^n (x_i^\top \hat{\theta} - x_i^\top \theta - \epsilon_i)^2 \right]$$

$$= \frac{1}{2n} \mathbb{E}_\epsilon \left[ \left\| X\theta - X\hat{\theta} \right\|_2^2 \right] - \frac{1}{n} \mathbb{E}_\epsilon \left[ \epsilon^\top (X\hat{\theta} - X\theta) \right] + \frac{1}{2n} \mathbb{E}_\epsilon \left[ \|\epsilon\|_2^2 \right]$$

$$= g_V(\hat{\theta}) - \frac{1}{n} \mathbb{E}_\epsilon \left[ \epsilon^\top (X\hat{\theta} - X\theta) \right] + \frac{\sigma^2}{2}$$

$$= g_V(\hat{\theta}) - \frac{1}{n} \mathbb{E}_\epsilon \left[ \sum_{i \in S} \epsilon_i (x_i^\top \hat{\theta} - x_i^\top \theta) \right] - \frac{1}{n} \mathbb{E}_\epsilon \left[ \sum_{i \in [n] \setminus S} \epsilon_i (x_i^\top \hat{\theta} - x_i^\top \theta) \right] + \frac{\sigma^2}{2}$$

Under the infinite data limit as $n \to \infty$, we have $\lim_{n \to \infty} \frac{1}{n} \mathbb{E}_\epsilon \left[ \sum_{i \in S} \epsilon_i (x_i^\top \hat{\theta} - x_i^\top \theta) \right] = 0$ since $S$ is a finite set. Since $\hat{\theta}$ only depends on $\epsilon_S$, the independence of $\hat{\theta}$ and $\epsilon_i$, $i \in [n] \setminus S$ can be established. Thus,

$$\mathbb{E}_\epsilon \left[ \sum_{i \in [n] \setminus S} \epsilon_i (x_i^\top \hat{\theta} - x_i^\top \theta) \right] = \sum_{i \in [n] \setminus S} \mathbb{E}_\epsilon [\epsilon_i] \mathbb{E}_\epsilon \left[ x_i^\top \hat{\theta} - x_i^\top \theta \right] = 0.$$

As a consequence, as $\lim_{n \to \infty} g(\hat{\theta}) - g_V(\hat{\theta}) - \frac{\sigma^2}{2} = 0$ for all $\hat{\theta} \in \Theta$.

$\square$

Note that the Proposition 4 does not imply that our algorithm performs the same steps when used with $g_V$ instead of $g$. It only means that the optimal solutions of the problems are converging to selections with the same quality in the infinite data limit.

**Bayesian V-Experimental Design.** Bayesian experimental designs [8] can be incorporated as well into our framework. In Bayesian modelling, the "true" parameter $\theta$ is not a fixed value, but instead a sample from a prior distribution $p(\theta)$ and hence a random variable. Consequently, upon taking into account the random nature of the coefficient vector we can find an appropriate inner and outer objectives.

**Proposition 5.** *Under Bayesian linear regression assumptions and where $\theta \sim \mathcal{N}(0, \lambda^{-1}I)$, let the outer objective*

$$g_V(\hat{\theta}) = \frac{1}{2n}\mathbb{E}_{\epsilon,\theta}\left[\left\|X\theta - X\hat{\theta}\right\|_2^2\right],$$

*where expectation is over the prior as well. Further, let the inner objective be equal to* (7) *with the same value of $\lambda$, then the overall objective simplifies to*

$$G(w) = \frac{1}{2n}\mathbf{Tr}\left(X\left(\frac{1}{\sigma^2}X^\top D(w)X + \lambda I\right)^{-1}X^\top\right). \tag{19}$$

*Proof.* Using the closed from in (8), and model assumptions, we see that $\hat{\theta} = (X_S^\top X_S + \lambda\sigma^2 I)^{-1}X_S^\top(X_S\theta + \epsilon_S)$. Plugging this in to the outer objective $g_V(\hat{\theta})$,

$$
\begin{aligned}
G(w) &= \frac{1}{2n}\mathbb{E}_{\epsilon,\theta}\left[\left\|X\theta - X\hat{\theta}\right\|_2^2\right] \\
&= \frac{1}{2n}\mathbb{E}_{\epsilon,\theta}\left[\left\|X((X_S^\top X_S + \lambda\sigma^2 I)^{-1}X_S^\top(X_S\theta + \epsilon_S) - \theta)\right\|_2^2\right] \\
&= \frac{1}{2n}\mathbb{E}_{\epsilon,\theta}\left[\left\|X(X_S^\top X_S + \lambda\sigma^2 I)^{-1}X_S^\top\epsilon_S - \sigma^2\lambda X(X_S^\top X_S + \lambda\sigma^2 I)^{-1}\theta\right\|_2^2\right] \\
&= \frac{1}{2n}\mathbb{E}_\theta\left[\left\|\lambda\sigma^2 X(X_S^\top X_S + \lambda\sigma^2 I)^{-1}\theta\right\|_2^2\right] + \frac{1}{2n}\mathbb{E}_\epsilon\left[\left\|X(X_S^\top X_S + \lambda\sigma^2 I)^{-1}X_S^\top\epsilon_S\right\|_2^2\right] \\
&= \frac{\sigma^2}{2n}\mathbf{Tr}\left(\lambda\sigma^2(X_S^\top X_S + \lambda\sigma^2 I)^{-1}X^\top X(X_S^\top X_S + \lambda\sigma^2 I)^{-1}\right) + \\
&\quad + \frac{\sigma^2}{2n}\mathbf{Tr}(X_S(X_S^\top X_S + \lambda\sigma^2 I)^{-1}X^\top X(X_S^\top X_S + \lambda\sigma^2 I)^{-1}X_S^\top) \\
&= \frac{\sigma^2}{2n}\mathbf{Tr}\left((X_S^\top X_S + \lambda\sigma^2 I)^{-1}X^\top X(X_S^\top X_S + \lambda\sigma^2 I)^{-1}\left(\lambda\sigma^2 I + X_S^\top X_S\right)\right) \\
&= \frac{\sigma^2}{2n}\mathbf{Tr}\left(X(X_S^\top X_S + \lambda\sigma^2 I)^{-1}X^\top\right) = \frac{\sigma^2}{2n}\mathbf{Tr}\left(X(X^\top D(w)X + \lambda\sigma^2 I)^{-1}X^\top\right)
\end{aligned}
$$

where we used that $\mathbb{E}_\epsilon[\epsilon] = 0$, and cyclic property of the trace, and the final results follows by rearranging.

$\square$

Similarly to the case of unregularized frequentist experimental design, in the infinite data limit, even the Bayesian objectives share the same optima. The difference here is that the true parameter is no longer a fixed value and we need to integrate over it using the prior.

**Proposition 6** (Infinite data limit). *Let $g_V$ be*

$$g_V(\hat{\theta}) = \frac{1}{2n}\mathbb{E}_{\epsilon,\theta}\left[\left\|X\theta - X\hat{\theta}\right\|_2^2\right]$$

*the Bayesian V-experimental design outer objective, and let the summarization objective be,*

$$g(\hat{\theta}) = \frac{1}{2n}\mathbb{E}_{\epsilon,\theta}\left[\sum_{i=1}^n(x_i^\top\hat{\theta} - y_i)^2\right].$$

*Under the same assumptions as in Proposition 4 and $\theta \sim \mathcal{N}(0, \lambda^{-1}I)$*

$$\lim_{n\to\infty} g(\hat\theta) - g_V(\hat\theta) - \frac{\sigma^2}{2} = 0, \quad \forall \hat\theta \in \Theta$$

.

*Proof.* The proof follows similarly as in Proposition 4. $\qquad\square$

**Weak-submodularity of Bayesian V-experimental design.** The greedy algorithm is known to perform well on A-experimental design due to the function $G(w)$ being weakly-submodular and monotone [6]. Cardinality-constrained maximization of such problems with greedy algorithm is known to find a solution which $1 - e^{-\gamma}$ [15] approximation to the optimal subset. The parameter $\gamma$ is known as weak submodularity ratio.

Following the analysis of [26], which derives the weak submodularity ratio of A-experimental design, we show that V-experimental design surprisingly has the same submodularity ratio as well, and hence we expect greedy strategy to perform well. We follow slightly different nomenclature than [26].

**Proposition 7.** *The function $R(w) = G(0) - G(w)$ as in (19) is non-negative, monotone and $\gamma$-weakly submodular function with*

$$\gamma = \left(1 + s^2 \frac{1}{\sigma^2\lambda}\right)^{-1}$$

*where $s = \max_{i\in\mathcal{D}} \|x_i\|_2$.*

*Proof.* We employ exactly the same proof technique as [26] relying on Sherman-Morison identity.

Let $A$ and $B$ be disjoint sets without loss of generality. Also, let $M_A := I\lambda + \frac{1}{\sigma^2} X_A X_A^\top$, which positive definite by definition. Following the line of proof in [26], it can be shown that a marginal gain of an element $e \in B$ is equal to

$$R(e|A) = \frac{x_e^\top M_A^{-1} X X^\top M_A^{-1} x_e}{\sigma^2 + x_e^\top M_A^{-1} x_e} \tag{20}$$

In order to derive the weak-submodularity ratio we need to lower bound,

$$\frac{\sum_{e\in B} R(e|A)}{R(B \cup A) - R(A)}. \tag{21}$$

Note the observation made by [26] that $\sigma^2 + x_e^\top M_A^{-1} x_e \le \sigma^2 + s^2\lambda^{-1}$ in their Equation 13.

$$\sum_{e\in B} R(e|A) = \sum_{e\in B} \frac{x_e^\top M_A^{-1} X X^\top M_A^{-1} x_e}{\sigma^2 + x_e^\top M_A^{-1} x_e} \tag{22}$$

$$= \sum_{e\in B} \frac{\mathbf{Tr}(x_e^\top M_A^{-1} X X^\top M_A^{-1} x_e)}{\sigma^2 + x_e^\top M_A^{-1} x_e} \tag{23}$$

$$\overset{\text{as above}}{\ge} \sum_{e\in B} \frac{\mathbf{Tr}(x_e^\top M_A^{-1} X X^\top M_A^{-1} x_e)}{\sigma^2 + s^2\lambda^{-1}} \tag{24}$$

$$= \frac{\mathbf{Tr}(X_B^\top M_A^{-1} X X^\top M_A^{-1} X_B)}{\sigma^2 + s^2\lambda^{-1}} \tag{25}$$

Now the denominator which is a more general version of (20)

$$R(B \cup A) - R(A) = \mathbf{Tr}\left((\sigma^2 I + X_B^\top M_A^{-1} X_B)^{-1} X_B^\top M_A^{-1} X X^\top M_A^{-1} X_B\right) \tag{26}$$

$$\le \frac{1}{\sigma^2} \mathbf{Tr}(X_B^\top M_A^{-1} X X^\top M_A^{-1} X_B) \tag{27}$$

where we used the fact that the $(\sigma^2 I + X_B^\top M_A^{-1} X_B) \succeq \sigma^2 I$. Note that the result follows by plugging (27) and (25) to the expression (21) and observing that the fraction can be simplified. The simplification finishes the proof. $\qquad\square$

**Lemma 8.** *Assume* $\|x_i\|_2 < L < \infty$ *for all* $i \in [n]$ *and* $w \in \mathbb{R}_+^n$ *s.t.* $\|w\|_2 < \infty$. *The function*

$$G(w) = \frac{1}{2n} \mathbf{Tr}\left( X \left( \frac{1}{\sigma^2} X^\top D(w) X + \lambda I \right)^{-1} X^\top \right)$$

*is convex and smooth in* $w$.

*Proof.* We will show that the Hessian of $G(w)$ is positive semi-definite (PSD) and that the maximum eigenvalue of the Hessian is bounded, which imply the convexity and smoothness of $G(w)$.

For the sake of brevity, we will work with the function $\hat{G}(w) = \mathbf{Tr}\left( X \left( X^\top D(w) X + \lambda \sigma^2 I \right)^{-1} X^\top \right)$ where $\frac{\sigma^2}{2n} \hat{G}(w) = G(w)$. Also, let us denote $F(w) = X^\top D(w) X + \lambda \sigma^2 I$ and $F^+(w) = \left( X^\top D(w) X + \lambda \sigma^2 I \right)^{-1}$ s.t. $F(w)F^+(w) = I$. First, we would like to calculate $\frac{\partial \hat{G}(w)}{\partial w_i}$, for which we will make use of directional derivatives:

$$
\begin{aligned}
D_v \hat{G}(w) &= \lim_{h \to 0} \frac{\hat{G}(w + hv) - \hat{G}(w)}{h} \\
&= \mathbf{Tr}\left( X \left( \lim_{h \to 0} \frac{F^+(w + hv) - F^+(w)}{h} \right) X^\top \right) \\
&= \mathbf{Tr}\left( X \left( \lim_{h \to 0} F^+(w + hv) \cdot \frac{F(w) - F(w + hv)}{h} \cdot F^+(w) \right) X^\top \right) \\
&\stackrel{\text{def. of } F}{=} -\mathbf{Tr}\left( X \left( \lim_{h \to 0} F^+(w + hv) \cdot \frac{\hbar X^\top D(v) X}{\hbar} \cdot F^+(w) \right) X^\top \right) \\
&= -\mathbf{Tr}\left( X F^+(w) X^\top D(v) X F^+(w) X^\top \right)
\end{aligned}
$$

In order to get $\frac{\partial \hat{G}(w)}{\partial w_i}$, we should choose as direction $v_i := (0, \ldots, 0, 1, 0, \ldots, 0)^\top$ where 1 is on the $i$-th position. Since $X^\top D(v_i) X = x_i x_i^\top$, we have that:

$$
\begin{aligned}
\frac{\partial \hat{G}(w)}{\partial w_i} = D_{v_i} \hat{G}(w) &= -\mathbf{Tr}\left( X F^+(w) x_i x_i^\top F^+(w) X^\top \right) \\
&\stackrel{\text{cyclic prop } \mathbf{Tr}}{=} -x_i^\top F^+(w) X^\top X F^+(w) x_i
\end{aligned}
$$

We will proceed similarly to get $\frac{\partial^2 \hat{G}(w)}{\partial w_j \partial w_i}$.

$$
\begin{aligned}
D_v \frac{\partial \hat{G}(w)}{\partial w_i} &= -x_i^\top \left( \lim_{h \to 0} \frac{F^+(w + hv) X^\top X F^+(w + hv) - F^+(w) X^\top X F^+(w)}{h} \right) x_i \\
&= -x_i^\top \left( \lim_{h \to 0} F^+(w + hv) \cdot \frac{X^\top X F^+(w + hv) F(w) - F(w + hv) F^+(w) X^\top X}{h} \cdot F^+(w) \right) x_i \\
&= x_i^\top \left( \lim_{h \to 0} F^+(w + hv) \cdot \frac{F(w + hv) F^+(w) X^\top X - X^\top X F^+(w + hv) F(w)}{h} \cdot F^+(w) \right) x_i
\end{aligned}
$$

Now, since,

$$F(w + hv)F^+(w) = (F(w) + hX^\top D(v)X)F^+(w) = I + hX^\top D(v)XF^+(w)$$
$$F^+(w + hv)F(w) = F^+(w + hv)(F(w + hv) - hX^\top D(v)X) = I - hF^+(w + hv)X^\top D(v)X$$

we have

$$
\begin{aligned}
D_v \frac{\partial \hat{G}(w)}{\partial w_i} &= x_i^\top F^+(w) \left( X^\top D(v) X F^+(w) X^\top X + X^\top X F^+(w) X^\top D(v) X \right) F^+(w) x_i \\
&= 2x_i^\top F^+(w) X^\top D(v) X F^+(w) X^\top X F^+(w) x_i
\end{aligned}
$$

Choosing $v_j$ as our directional derivative, we have:

$$\frac{\partial^2 \hat{G}(w)}{\partial w_j \partial w_i} = D_{v_j} \frac{\partial \hat{G}(w)}{\partial w_i} = 2 \left( x_i^\top F^+(w) x_j \right) \left( x_j^\top F^+(w) X^\top X F^+(w) x_i \right)$$

$$= 2 \left( x_j^\top F^+(w) x_i \right) \left( x_j^\top F^+(w) X^\top X F^+(w) x_i \right)$$

from which we can see that we can write the Hessian of $\hat{G}(w)$ in matrix form as:

$$\nabla_w^2 \hat{G}(w) = 2 \left( X F^+(w) X^\top \right) \circ \left( X F^+(w) X^\top X F^+(w) X^\top \right)$$

where $\circ$ denotes the Hadamard product. Since $F^+(w)$ is PSD it immediately follows that $X F^+(w) X^\top$ and $X F^+(w) X^\top X F^+(w) X^\top$ are PSD. Since the Hadamard product of two PSD matrices is PSD due to the *Schur product theorem*, it follows that the Hessian $\nabla_w^2 \hat{G}(w)$ is PSD and thus $G(w)$ is convex.

As for smoothness, we need the largest eigenvalue of the Hessian to be bounded:

$$\lambda_{\max}(\nabla_w^2 \hat{G}(w)) \quad \leq \quad \mathbf{Tr}(\nabla_w^2 \hat{G}(w)) = 2 \sum_{i=1}^{n} \left( X F^+(w) X^\top \right)_{ii} \left( X F^+(w) X^\top X F^+(w) X^\top \right)_{ii}$$

$$= \quad 2 \sum_{i=1}^{n} \left( x_i^\top F^+(w) x_i \right) \left( x_i^\top F^+(w) X^\top X F^+(w) x_i \right)$$

$$= \quad 2 \sum_{i=1}^{n} \left( x_i^\top F^+(w) x_i \right) \left\| X F^+(w) x_i \right\|_2^2$$

$$\overset{\text{Rayleigh q.}}{\leq} \quad 2 \sum_{i=1}^{n} \lambda_{\max}(F^+(w)) \left\| x_i \right\|_2^2 \left\| X F^+(w) x_i \right\|_2^2$$

$$\leq \quad 2 \sum_{i=1}^{n} \lambda_{\max}(F^+(w)) \left\| x_i \right\|_2^2 \left\| X \right\|_2^2 \left\| F^+(w) \right\|_2^2 \left\| x_i \right\|_2^2$$

$$= \quad 2 \lambda_{\max}^3(F^+(w)) \left\| X \right\|_2^2 \sum_{i=1}^{n} \left\| x_i \right\|_2^4 \leq \frac{2}{\lambda^3 \sigma^6} \left\| X \right\|_2^2 \sum_{i=1}^{n} \left\| x_i \right\|_2^4$$

$$\leq \quad \frac{2}{\lambda^3 \sigma^6} \left\| X \right\|_F^2 \, n L^4 \leq \frac{2 n^2 L^6}{\lambda^3 \sigma^6}$$

Thus $G$ is $\frac{n L^6}{\lambda^3 \sigma^4}$-smooth.

$\qquad\qquad\qquad\qquad\qquad\qquad\qquad\qquad\qquad\qquad\qquad\qquad\qquad\qquad\qquad\qquad\qquad\qquad\square$

# B    Connections to Influence Functions.

We show that our approach is also related to incremental subset selection via influence functions. Let us consider the influence of the $k$-th point on the outer objective. Suppose that we have already selected the subset $S$ and found the corresponding optimal weights $w_S^*$. Then, the influence of point $k$ on the outer objective is

$$\mathcal{I}(k) := \left. -\frac{\partial \sum_{i=1}^{n} \ell_i(\theta^*)}{\partial \varepsilon} \right|_{\varepsilon = 0}$$

$$\text{s.t.} \quad \theta^* = \arg\min_{\theta} \sum_{i=1}^{n} w_{S,i}^* \, \ell_i(\theta) + \varepsilon \boldsymbol{\ell_k}(\boldsymbol{\theta}).$$

**Proposition 9.** *Under twice differentiability and strict convexity of the inner loss, $\arg\max_k \mathcal{I}(k)$ corresponds to the selection rule in Equation (5).*

*Proof.* Following [34] and using the result of [14], the empirical influence function at $k$ is

$$\left.\frac{\partial \theta^*}{\partial \varepsilon}\right|_{\varepsilon=0} = -\left(\frac{\partial^2 \sum_{i=1}^n w_{S,i}^* \ell_i(\theta^*)}{\partial\theta\partial\theta^\top}\right)^{-1} \nabla_\theta \ell_k(\theta^*). \tag{28}$$

Now, we use the chain rule for $\mathcal{I}(k)$:

$$\begin{aligned}
\mathcal{I}(k) &= -\left.\frac{\partial \sum_{i=1}^n \ell_i(\theta^*)}{\partial\varepsilon}\right|_{\varepsilon=0} \\
&= -\left(\nabla_\theta \sum_{i=1}^n \ell_i(\theta^*)\right)^\top \left.\frac{\partial\theta^*}{\partial\varepsilon}\right|_{\varepsilon=0} \\
&\overset{\text{Eq. 28}}{=} \nabla_\theta \ell_k(\theta^*)^\top \left(\frac{\partial^2 \sum_{i=1}^n w_{S,i}^* \ell_i(\theta^*)}{\partial\theta\partial\theta^\top}\right)^{-1} \nabla_\theta \sum_{i=1}^n \ell_i(\theta^*).
\end{aligned}$$

$\square$

## C Merge-reduce Streaming Coreset Construction

---
**Algorithm 2** streaming_coreset
---
**Input:** stream $S$, number of slots $s$, $\beta$
buffer $= [\,]$
**for** $\mathcal{X}_t$ in stream $S$ **do**
  $\mathcal{C}_t = $ construct_coreset$(\mathcal{X}_t)$
  buffer.append$((\mathcal{C}_t, \beta))$
  **if** buffer.size $> s$ **then**
    $k = $ select_index(buffer)
    $\mathcal{C}' = $ construct_coreset$((\mathcal{C}_k, \beta_k), (\mathcal{C}_{k+1}, \beta_{k+1}))$
    $\beta' = \beta_k + \beta_{k+1}$
    delete buffer$[k+1]$
    buffer$[k] = (\mathcal{C}', \beta')$
  **end if**
**end for**
---

---
**Algorithm 3** select_index
---
**Input:** buffer $= [(C_1, \beta_1), \dots, (C_{s+1}, \beta_{s+1})]$ containing an extra slot
**if** $s == 1$ or $\beta_{s-1} > \beta_s$ **then**
  **return** $s$
**else**
  $k = \arg\min_{i\in[1,\dots,s]} (\beta_i == \beta_{i+1})$
  **return** $k$
**end if**
---

## D Continual Learning and Streaming Experiments

We conduct an extensive study of several selection methods for the replay memory in the continual learning setup. We compare the following methods:

- Training w/o replay: train after each task without replay memory. Demonstrates how catastrophic forgetting occurs.
- Uniform sampling / per task coreset: the network is only trained on the points in the replay memory with the different selection methods.

Table 5: Continual learning with replay memory size of 100 for versions of MNIST and 200 for CIFAR-10. We report the average test accuracy over the tasks with one standard deviation over 5 runs with different random seeds. Our coreset construction performs among the best.

| Method | PermMNIST | SplitMNIST | SplitCIFAR-10 |
|---|---|---|---|
| Training w/o replay | $73.82 \pm 0.49$ | $19.90 \pm 0.03$ | $19.95 \pm 0.02$ |
| Uniform sampling, train at the end | $34.31 \pm 1.37$ | $86.09 \pm 1.84$ | $28.31 \pm 0.94$ |
| Per task coreset, train at the end | $46.22 \pm 0.45$ | $92.53 \pm 0.53$ | $31.01 \pm 1.16$ |
| Uniform sampling | $78.46 \pm 0.40$ | $92.80 \pm 0.79$ | $36.20 \pm 3.19$ |
| $k$-means of features | $78.34 \pm 0.49$ | $93.40 \pm 0.56$ | $33.41 \pm 2.48$ |
| $k$-means of embeddings | $78.84 \pm 0.82$ | $93.96 \pm 0.48$ | $36.91 \pm 2.42$ |
| $k$-means of grads | $76.71 \pm 0.68$ | $87.26 \pm 4.08$ | $27.92 \pm 3.66$ |
| $k$-center of features | $77.32 \pm 0.47$ | $93.16 \pm 0.96$ | $32.77 \pm 1.62$ |
| $k$-center of embeddings | $78.57 \pm 0.58$ | $93.84 \pm 0.78$ | $36.91 \pm 2.42$ |
| $k$-center of grads | $77.57 \pm 1.12$ | $88.76 \pm 1.36$ | $27.01 \pm 0.96$ |
| Gradient matching | $78.00 \pm 0.57$ | $92.36 \pm 1.17$ | $35.69 \pm 2.61$ |
| Max entropy samples | $77.13 \pm 0.63$ | $91.30 \pm 2.77$ | $27.00 \pm 1.83$ |
| Hardest samples | $76.79 \pm 0.55$ | $89.62 \pm 1.23$ | $28.10 \pm 1.79$ |
| FRCL's selection | $78.01 \pm 0.44$ | $91.96 \pm 1.75$ | $34.50 \pm 1.29$ |
| iCaRL's selection | $79.68 \pm 0.41$ | $93.99 \pm 0.39$ | $34.52 \pm 1.62$ |
| Coreset | $79.26 \pm 0.43$ | $95.87 \pm 0.20$ | $37.60 \pm 2.41$ |

- Training per task with uniform sampling / coreset: the network is trained after each task and regularized with the loss on the samples in the replay memory chosen as a uniform sample / coreset per task.
- $k$-means / $k$-center in feature / embedding / gradient space: the per-task selection retains points in the replay memory that are generated by the $k$-means++ [4] / greedy $k$-center algorithm, where the clustering is done either in the original feature space, in the last layer embedding of the neural network, or in the space of the gradient with respect to the last layer (after training on the respective task). The points that are the cluster centers in the different spaces are the ones chosen to be saved in the memory. Note that the $k$-center summarization in the last layer embedding space is the coreset method proposed for active learning by [49].
- Hardest / max-entropy samples per task: the saved points have the highest loss after training on each task / have the highest uncertainty (as measured by the entropy of the prediction). Such selection strategies are used among others by [12] and [1].
- Training per task with FRCL's / iCaRL's selection: the points per task are selected by FRCL's inducing point selection [53], where the kernel is chosen as the linear kernel over the last layer embeddings / iCaRL's selection (Algorithm 4 in [46]) performed in the normalized embedding space.
- Gradient matching per task: same as iCaRL's selection, but in the space of the gradient with respect to the last layer and jointly over all classes. This is a variant of Hilbert coreset [7] with equal weights, where the Hilbert space norm is chosen to be the squared 2-norm difference of loss gradients with respect to the last layer.

We report the results in Table 5. We note that while several methods outperform uniform sampling on some datasets, only our method is consistently outperforming it on all datasets. We also conduct a study on the effect of the replay memory size shown in Table 6.

Table 6: Replay memory size study on SplitMNIST. Our method offers bigger improvements with smaller memory sizes.

| Method / Memory size | 50 | 100 | 200 |
|---|---|---|---|
| CL uniform sampling | $85.23 \pm 1.84$ | $92.80 \pm 0.79$ | $95.08 \pm 0.30$ |
| CL coreset | $91.79 \pm 0.71$ | $95.87 \pm 0.20$ | $97.06 \pm 0.30$ |
| Streaming reservoir sampling | $83.90 \pm 3.18$ | $90.72 \pm 0.97$ | $94.12 \pm 0.61$ |
| Streaming coreset | $85.68 \pm 2.05$ | $92.59 \pm 1.20$ | $95.64 \pm 0.68$ |

**RBF kernel as proxy.**   In our experiments we used the Neural Tangent Kernel as proxy. It turns out that on the datasets derived from MNIST simpler kernels such as RBF are also good proxy choices. To illustrate this, we repeat the continual learning and streaming experiments and report the results in Table 7. For the RBF kernel $k(x,y) = \exp(-\gamma \left\| x - y \right\|_2^2)$ we set $\gamma = 5 \cdot 10^{-4}$. While the RBF kernel is a good proxy for these datasets, it fails on harder datasets such as CIFAR-10.

Table 7: RBF vs CNTK proxies.

|  | Method | PermMNIST | SplitMNIST |
|---|---|---|---|
| CL | Coreset CNTK | $79.26 \pm 0.43$ | $95.87 \pm 0.20$ |
| | Coreset RBF | $79.89 \pm 0.87$ | $96.18 \pm 0.33$ |
| VCL | Coreset CNTK | $86.18 \pm 0.21$ | $84.66 \pm 0.63$ |
| | Coreset RBF | $86.13 \pm 0.31$ | $82.37 \pm 1.40$ |
| Str. | Coreset CNTK | $74.44 \pm 0.52$ | $92.59 \pm 1.20$ |
| | Coreset RBF | $75.93 \pm 0.59$ | $92.61 \pm 0.83$ |

**Inspecting the coreset.**   We provide further insights for the large gains obtained by our method by inspecting the resulting coreset when summarizing MNIST in Figure 5. We can notice that in each step, the method selects the sample that has the potential to increase the accuracy by the largest amount: the first 10 samples (first row) are picked from different classes, after which the method diversifies within the classes. Even though the dataset is balanced, our method chooses more samples from more diverse classes, e.g., it picks twice as many 8s than 1s.

Figure 5: Coreset of size 40 of MNIST produced by our method: samples are diverse within class, and harder classes are represented with more samples.

# E   Speedups, Data Preprocessing, Architectures and Hyperparameters

**Speedups.**   We now discuss several speedups. Since the number of parameters in the inner optimization problem in Eq. 6 is small, we can use quasi-Newton methods such as L-BFGS [36] for faster convergence. For the outer level optimization we use Adam [31]. After a step on the outer level, we can reuse the result from the last inner iteration to restart the optimization of the inner variable. For additional speedup in the streaming and continual learning setting, we use binary weights and consequently do not perform weight optimization.

**Bilevel optimization.**   When the inner optimization problem cannot be solved in closed form, we solve it using L-BFGS using step size 0.25, 200 iterations and $10^{-5}$ tolerance on first order optimality. For the inner optimization problem we report the best result obtained for $\lambda \in \{10^{-6}, 10^{-5}, 10^{-4}, 10^{-3}, 10^{-2}\}$. The number of conjugate gradient steps per approximate inverse Hessian-vector product calculation is set to 50.

For the outer level we use Adam, with learning rate 0.05 and 10 iterations for the CNN on MNIST experiment, and with learning rate 0.02 and 200 iterations for the KRR with CNTK on CIFAR-10 experiment — here we can afford more outer iterations as the inner problem can be solved in closed form. When applying the gradient on weights, we ensure the positivity of weights by projection. In continual learning and streaming we use unweighted samples, so we do not perform outer iterations. When applying our incremental selection of points, we select the best candidate out of a random sample of 200.

**Data preprocessing.**   For all datasets, we standardize the pixel values per channel for each dataset separately. For the CIFAR-10 streaming experiment we perform the standard data augmentations of random cropping and horizontal flipping.

**Training details.**   In the continual learning experiments, we train our networks for 400 epochs using Adam with step size $5 \cdot 10^{-4}$ after each task. The loss at each step consists of the loss on a minibatch of size 256 of the current tasks and loss on the replay memory scaled by $\beta$. For streaming, we train our networks for 40 gradient descent steps using Adam with step size $5 \cdot 10^{-4}$ after each batch. For [2], we use a streaming batch size of 10 for better performance, as indicated in Section 2.4 of the supplementary materials of [2].

**Hyperparameter selection in continual learning and streaming.**   The replay memory regularization strength $\beta$ was tuned separately for each method from the set $\{0.01, 0.1, 1, 10, 100, 1000\}$ and the best result on the test set was reported.

**Computational resources.**   The neural networks are trained on a single GeForce GTX 1080 Ti. (C)NTK kernels are calculated on the same GPU. The coreset generation is performed with a single CPU thread.