[Reviews · NeurIPS 2020]

Review 1

Summary and Contributions: This work proposes generating a weighted coreset as the answer to a bilevel optimization problem. Building on techniques of Refs [23], [38] authors propose first-order optimization method to obtain the coreset. The authors connect their approach to optimal experimental design, and show applicability of first-order methods as the overall objective in that framework becomes convex over the weights. To finally be able to apply the approach for learning parameters of a neural network, they consider a proxy model for the network that can be written explicitly using a kernel function. The authors then apply their approach to experiments on dataset summarization and continual learning.

Strengths: The claims in the paper are theoretically sound and build upon prior work in a non-trivial way. For example, when the inner optimization is a weighted and regularized least squares regression and the outer objective is sum of squared errors, the overall objective is shown to be convex in the choice of weights. The experimental results show significant improvement, and the considered problem is relevant.

Weaknesses: In the empirical evaluation, the interaction between picking a good coreset vs a good kernel is unclear to me. For example, for m > 200 in fig 2c, it seems that picking a subset uniformly and then optimizing weights for CNTK gives better accuracy than the proposed approach applied using RBF kernel.

Correctness: I have briefly checked the proof of Lemma 8 in the appendix and that seemed correct to me.

Clarity: The paper is generally written ok, some times a bit hard to follow due to frequent references to the appendix. It would be good to have some of the lemma statements in the main body of the paper.

Relation to Prior Work: There is adequate discussion about related work, and they have been extensively compared against the proposed approach in the experiment section.

Reproducibility: Yes

Additional Feedback: After reading author response: I thank the authors for the clarification, and I don't need to change my score.


Review 2

Summary and Contributions: The authors propose a new method, inspired by coresets, to do data summarization in the streaming and continual learning setting that is applicable to a broad class of machine learning algorithms, including neural networks. In fact, it is applicable to any ML algorithm that is based on empirical risk minimization. The authors pose the coreset selection problem as a bilevel optimization problem and propose to solve it via a greedy selection mechanism using matching pursuit. Additionally, to achieve practical computation times for neural networks the authors propose to use a proxy model (RHKS) for solving the inner optimization problem in the case of neural networks to avoid re-training the network for each evaluation of the inner objective.

Strengths: Overall, this work is both novel, insightful, and really easy to follow. I really appreciated the rigor with which the authors derived their coreset algorithm starting from a very general problem formulation (c.f. bi-level optimization) and intuitively explained each step along the way until they arrive at their final algorithm. Given the generality of the formulation future work will hopefully be available to build upon their framework and improve upon the individual approximation steps to potentially design even better algorithms. The algorithm seems to combine advances and state-of-the-art algorithms from different fields and I commend the authors for combining these advances into an interesting, computationally-tractable algorithm for coresets. The extension from the offline scenario to the online scenario (streaming and continual) was well explained and intuitive to follow.

Weaknesses: My major concern is related to the scope of the experiments. The authors throughout the paper claim that their method is the new de-facto state-of-the-art method for continual learning and streaming, however, the experimental evidence lags behind those claims in my opinion. I don’t think experiments on MNIST alone are enough to show superiority of their method. While there was one CIFAR experiment with ResNets, I believe an array of modern, large-scale architectures trained on CIFAR should be the standard evaluation for all their experiments. Adding ImageNet experiments on top of that would of course be a bonus but not necessarily required in my opinion, especially because it would probably require another approximation step in order to achieve practical computation times. While I appreciate some of the theoretical results that are presented in the main part of the body, they seem hardly applicable to neural networks. Given that the main motivation of the authors is coresets for NNs, I would appreciate some additional theoretical results pertaining to NNs. I understand that it might be too much to ask since it might be very hard to get any kind of bounds though. So it would be nice to at least have some additional discussion about the connection (or non-existing connection) of the theoretical results to NNs. In the CIFAR experiments it seems that the CNTK kernel was based on a different neural network than the actual neural network which was trained. Is that a necessary step because the ResNet18 used for the CIFAR experiment is too large to even compute the CNTK kernel? If so, I think that would merit some discussion much earlier in the paper since that significantly narrows the applicability of the proposed method. As it stands now it seems you need a proxy to get it to work with NNs at all and then in order to get it to work with large-scale NNs you need another small NN in order to evaluate the proxy, so it’s like a “proxy of a proxy”. That seems like quite a big step which is hard to judge from just one experiment and no further discussion. In summary, I can see two avenues for the authors to improve their paper, in which case it would be a clear accept for me: 1. add additional experiments for CIFAR and clarify whether and how their proposed method works for larger NNs that are usually used in conjunction with CIFAR or whether they require an additional approximation step (such as basing the kernel of the proxy model on a smaller NN as was done in their only CIFAR experiment). The additional approximation step would have to be justified though in my opinion. Given the novelty of their method, I would be happy to raise my score even if their method doesn’t significantly outperform existing methods. 2. focusing on the more general setting, which is that their method is more broadly applicable to other ML algorithms as well and not specific to NNs. In that case, I would like to see evaluations on a broader range of ML algorithms to see how their algorithm performs against the respective state-of-the-art for those algorithms. In that case, they could provide a unified framework which is really exciting and interesting on its own. Then I would also be fine with MNIST-only experiments for NNs given that the authors specify that they cannot claim state-of-the-art for all neural networks since their experiments are limited to MNIST NNs.

Correctness: I haven’t checked the proofs of the paper in detail but the claims and methods are sound and the provided explanations for the derivations are helpful in verifying the correctness of the method.

Clarity: The paper is very well-written and intuitive to follow.

Relation to Prior Work: I think their discussion on related work is extensive and puts their work in an appropriate context to previous work. However, I think that the authors could be more straightforward in the introduction (maybe through a list of contributions) that their work does not introduce a new algorithm per-se but rather uses existing techniques like matching pursuit [38] and the merge-and-reduce framework [38] which they apply within a novel context. In my opinion, that doesn’t degrade the novelty of their work but would help clarify early on what their actual contribution is.

Reproducibility: Yes

Additional Feedback: * Equations (3) and (5) seem correct but aren’t straightforward, in my opinion. Re-deriving them in the appendix with a few extra steps and explanation could be really useful for the interested reader. * I am not entirely sure how the equivalence with respect to the influence function contributes to the understanding of the method. It seems interesting but I would love to see further clarification of that connection in order to help me contextualize the value of it. * The choice of the kernel seems to be central for the empirical evaluations. I would love to see a longer discussion on the choice of kernels. I am not familiar with the details of [29] that presumably introduces the NTK kernel but it seems to be too central to this paper as to be omitted from the main body. * small typo on line 178: “results” —> “results in” ========================================================= UPDATE AFTER REBUTTAL ========================================================= I think the rebuttal was solid and I appreciated that the authors provided additional experiments. I think it’s good that they got their method to work on ResNets with only one proxy instead of a “proxy of a proxy”. That seems to simplify the deployment of the method since now a potential user of the method is only faced with one heuristic choice (i.e. picking the kernel) instead of two. I think the increased run time is okay in this case. However, what still worries me even after their rebuttal is that since NNs are such a empirical field, I still don’t have a good grasp of the limitations of their method. What happens with larger coresets? What about not trying to use a kernel at all (even if it takes too long) as suggested by R5 in their review? Given that their paper and rebuttal is mainly focused on NNs they seem to target NN applications in particular. So it would be important to understand the limitations. Overall though, I think it’s an interesting paper with a decent framework. And data summarization technique are an important field. And while many papers target specific applications, it’s encouraging to see a framework that works well across multiple settings. Notwithstanding my criticism with regards to understanding the limitations, I am increasing my score to 6.


Review 3

Summary and Contributions: The paper proposes a method for selecting corsets that are representative of a given dataset. The proposed approach follows a bilevel optimization solution of the coreset selection and the parameters minimizing the loss. Theoretical guarantees were shown in the case of convex problems, however, this is not the case of neural networks. When speaking of neural networks and large datasets, the proposed solutions involves prohibited (complex) operations such as inverting the hessian in each iteration, this is (to an extent) overcome by using an approximate model during the coreset selection e.g. neural tangent kernel (NTK) or RBF. Results on learning on coreset and continual learning both task based and online streaming shows the improvements of the proposed solution. =============Post Rebuttal============= The rebuttal has clarified some of my concerns however, I am still not happy with the experimental evaluation and why the authors didn't compare with GSS on Mnist Imbalanced using GSS setting as I requested. I am keeping my score as the idea of work is novel and the problem is crucial for continual learning.

Strengths: A new coreset selection method based on cone constrained generalized matching pursuit. An approximated solution for neural network based on kernels. An extension to incremental setting which makes the solution applicable to the problem of continual learning where replay buffer is a very important factor especially in the more challenging streaming scenario.

Weaknesses: Although the approach is theoretically grounded but a series of approximation and heuristic is applied to make the solution applicable to neural network yet by further approximating the network with a kernel. My main concern the the computation cost of the proposed method, it is discussed briefly in the paper but it is very hard to estimate how does it compare to existing approaches. Only a very small coreset size is considered which also might be due to complexity burden. It is mentioned that the coreset selection takes half time the training but this might be I would like to see a time cost comparison with other methods. The imbalanced setting is very close to that described in [2](table 4) , however in this paper only one imbalanced sequence is shown and by only comparing to Reservoir. I would like to see a comparison under the same setting of [2], I believe the code of the experiment is available, please also include a time cost comparison.

Correctness: yes claims are justified and empirical validation seems sound.

Clarity: yes but a lot of details are left to supplementary which makes the paper hard to follow. One general algorithm for the streaming setting could be helpful.

Relation to Prior Work: yes.

Reproducibility: Yes

Additional Feedback:


Review 4

Summary and Contributions: The paper develops a greedy method based on corsets for data continual learning, i.e., finding a summarization of the data to be used for future training/inference. The algorithm is tweaked to be used for continual learning where the model is a Neural Network (NN). Since the original formulation requires computing the Hessian of an over-parameterized model in NN, the algorithm resorts to kernel approximation to the NN. Paper uses empirical justification to prove that their algorithm is superior to some (fairly easy) baselines.

Strengths: Brining the concept of corsets to the Continual Learning problem is interesting. Data summarization is in general an important problem in machine learning and creating the connection with other related areas is important. While bringing some of the fundamental ideas of bilevel optimization from [52] limits the significance of the work, the connections remain interesting.

Weaknesses: The main weakness of the paper is reliance on mostly empirical evidence in justifying the proposed scheme. While empirical results are fine, they are compared with fairly easy baselines that are not directly designed for the problem at hand. As an example, Uniform and Reservoir Sampling are simple and data-agnostic random sampling strategies and while serve as a good control might not be very informative in judging the quality of the results in the paper. One question come to mind is if other simple but smarter baseline can be devised, e.g., a simple baseline for the class-imbalance problem improving the Reservoir sampling might be the one that samples from every class not uniformly but propositional to class frequency? (A Count Sketch can be painted in the streaming case to record the number of images in each class.)

Correctness: Claims and mythologies are reasonable. However, since most of the justifications are empirical that are hard to assess. Some other important empirical justifications are also lacking. For example, an important question might be how much of an approximation would be to use corsets with RBF/CNTK compared to using the original NN? What is the best way to find these kernels? Are these kernels feasible for real-world massive networks such as RESNET? Also in the case of the non-convex NN do we know how much of an approximation the greedy scheme introduces? Does the (near)submodularity still hold? Another question is on the experimental data. While some are novel and interesting, they are very small to be representative of a streaming problem: Hundreds to thousands of MNIST-sized data. I would have appreciated some experiments on real-world streaming problems to really assess the importance of the algorithm for Continual Learning.

Clarity: The paper is well written. I was able to follow most of the text. The only section that would suggest improving is section 3.4 on Relation to Influence Functions which I think it can be improved with more elaboration and why this is important and how it blends with the main message of the paper.

Relation to Prior Work: Connections to other works has been cited and descried well. *** after rebuttal. I have read the response letter carefully. I suggested adding discussions on the limitations of the work especially around NN given the approximations made in the algorithm.

Reproducibility: Yes

Additional Feedback:

[Author Response · NeurIPS 2020]

We thank the reviewers for their time and insightful feedback. We now address the concerns. **C** - Concern, **R** - Response

**(*) To all reviewers. C: Dataset choices and the scale of the experiments. R**: While our proposed method is more
generally applicable, we showcased it for experience replay-based continual learning and streaming, where the idea of
coresets is largely under-explored. In these scenarios, the *standard datasets* used for evaluation are *SplitMNIST and*
*PermMNIST* and their variations, see [44, 16, 2, 9], and the standard architectures are fully-connected nets or simple
CNNs. Commonly, the experience replay size is also restricted to a few samples to emulate a memory-constrained
environment. Thus, we decided to use the *same setup* also in our paper to *conform to the practices* in the field and to be
easily comparable to competing methods, e.g., VCL. For the final version of paper we will add more summarization and
continual learning experiments of CIFAR. **C: Proxy kernels and ResNets. R:** With recent libraries (e.g., [45] used in
our work), the CNTK can be *calculated efficiently* even for ResNets. To prove this point, we *repeated our imbalanced*
*streaming* experiment with the CNTK of *ResNet-18* and report the results in Table 1. This kernel provides an accuracy
improvement $1.7\%$ over the kernel used in the submission at the expense of moderately increased runtime, as shown in
Table 2. However, CNTK is only one particular proxy choice. For the revised paper, we will expand our discussion and
results on the proxy choices. **C: Theoretical guarantees. R:** We proved in the paper that with L2 loss and infinitely
wide neural networks our coreset construction provides convergence of order 1/T as per Theorem 1. However, with
other losses and finite-width neural networks both the bilevel coreset and neural network optimization problems become
    NP-hard in general, thus convergence guarantees are not easily obtainable. We will clarify this in the paper.

Table 1: New imbalanced CIFAR-10 streaming results.

|  | SplitCIFAR-10 |
|---|---|
| **Coreset + SimpleCNTK (original)** | $32.30 \pm 0.84$ |
| **Coreset + ResNet-18CNTK** | $33.98 \pm 1.44$ |
| **OCL [A]** | $32.25 \pm 1.69$ |

Table 2: Runtimes for generating a coreset of size 100 out of 1000 points with CNTK.

|  | SimpleCNTK | ResNet-18CNTK |
|---|---|---|
| **Coreset gen** | 57.3 s | 63.1 s |
| **Kernel calc** | 6.3 s | 56.2 s |

**Reviewer #1. C: Uniformly sampled, weight-optimized CNTK summary performs better than RBF coreset. R**:
This is indeed the case, since the CNTK is better suited for images than the RBF kernel, as it takes into account the
inductive biases of locality and translation invariance. Thus, the performance gap is due to the kernel choice rather than
the bilevel optimization scheme. For choosing the right kernel, we opt in the paper for choosing the CNTK equivalent
to a neural network that works well for the specific problem.

**Reviewer #2. C: Dataset choices (mostly MNIST) and further theoretical guarantees. R:** Please see (*). **C:**
**Evaluation with ResNets and CIFAR-10. R:** Please see (*). You correctly observe that we used "proxy of a proxy" in
the imbalanced streaming experiment. We used it for keeping the coreset generation time the lowest possible, while still
achieving good results. In our new CIFAR-10 experiments shown in Table 1, the CNTK corresponding to ResNet-18
provides an additional 1.7% compared to the SimpleCNTK proxy reported in the paper, but is 50 s slower in generating a
coreset of size 100 (Table 2). We have also performed the continual learning experiments on SplitCIFAR with ResNet-18
and ResNet-18CNTK as proxy, and observe a $1.9\%$ improvement with coresets over uniform sampling; we will report
the rest of the results and provide a more detailed discussion on the proxy choices in the final version of our paper.

**Reviewer #4. C: Coreset generation time burden. R:** Please see (*). We restricted the coreset sizes to the order
of hundreds as this is the standard evaluation practice in the continual learning literature. In this setting, our coreset
construction time is lower than the training time, thus our method is at most twice as slow as the fastest competing
method. We will provide wall-clock time measurements for summary generation sizes of the different methods in the
final version of the paper. **C: Imbalanced Streaming comparison to [2]. R:** Similarly as reported in the streaming
setting (line 302), we did perform such an experiment, but we were not able to tune the method of [2] to outperform
reservoir sampling - this is in line with the observation of the authors in [2] that their method requires a summary of
size at least 1k to work on CIFAR-10. We will include this observation together with the time comparison.

**Reviewer #5. C: Hardness of the baselines, CNTK choice, ResNets and further theoretical guarantees. R:** Please
see (*) and the new ResNet results in Table 1. **C: More challenging imbalanced streaming baselines. R:**. You
suggest to improve reservoir sampling by a method "that samples from every class not uniformly but propositional to
class frequency" - we note that this is *exactly* what reservoir sampling is doing on an imbalanced stream, i.e., keeps
more samples from more representative classes, thus under-representing minority classes. We believe you intended
to propose a class-balancing version of reservoir sampling similar to the Algorithm 1 of the concurrent work of [A].
We implemented this strategy designed for imbalanced streaming and report the results in Table 1, where we find it to
match the performance of our method with the SimpleCNTK kernel, and slightly underperforming compared to the
ResNet-18CNTK. We note that our method is general, and was not purposefully designed for imbalanced streams.

[A] Chrysakis et al. Online Continual Learning from Imbalanced Data, ICML 2020

[Meta-Review · NeurIPS 2020]

Authors propose a bilevel optimization based method for selecting coresets for continual learning and online applications. All reviewers agree on the merits of the submission. The raised issues are largely minor except the criticism on the experimental setup. I think the empirical study can be improved but the weakness is a rather acceptable one as continual learning is a new and upcoming topic without established baselines and empirical settings. Hence, many (possibly most) continual learning uses different empirical setups. Hence, I believe this is an acceptable weakness. Issues to fix in the camera ready: Authors should extend the discussion on the limitations of the work for neural networks.